# Reference intervals of common clinical biochemistry analytes in young Nigerian adults

**Ojor Ayemoba**[1]*, **Nathan Okeji**[1], **Nurudeen Hussain**[1], **Tahir Umar**[1], **Anthony Ajemba-Life**[1], **Terfa Kene**[2], **Uchechukwu Edom**[1], **Ikechukwu Ogueri**[1], **Goodluck Nwagbara**[1], **Inalegwu Ochai**[1], **Usman Adekanye**[1], **Ikenna Onoh**[3]

**1** Ministry of Defence Health Implementation Programme, Abuja, Nigeria, **2** Ave Health Sense Ltd, Abuja, Nigeria, **3** Nigerian Field Epidemiology and Laboratory Training Programme, Abuja, Nigeria

* orayemoba@yahoo.com

**Data Availability Statement:** All relevant data are within the manuscript and its Supporting information files.

## Abstract

### Background

Reference intervals are assessment tools for interpretation of clinical test results. These intervals describe the dispersion of test parameter values of apparently healthy persons in defined populations as health status indicators. Using reference intervals obtained and validated in populations outside the geographical region of derivation for medical decision-making may impact negatively on clinical interpretation and patient management. Many countries have established their reference values, current studies on these data for Nigeria are however scarce. Determination of clinical biochemistry reference intervals for young Nigerian adults which is of particular importance in routine clinical management and conduct of clinical trials in response to existing and emerging diseases will add significantly to the existing body of knowledge.

### Objective

The objective was to establish reference intervals for 24biochemistry analytes among Nigerians aged 18 to 26 years.

### Methods

This was a cross-sectional study among 7,797 consenting male and female military applicants aged 18 to 26 years from 37 States of Nigeria. It was a total study among volunteers for military service. Blood samples were collected and subjected to serological testing for HIV-1 and 2, hepatitis-B, malaria, pregnancy and haematuria to restrict our study population to apparently healthy participants. Biochemical assays were performed on 6,169 participant samples that met the inclusion criteria. Generated data was entered into MS Excel® and exported into SPSS® software version 16 for analysis. Statistical tools used were frequencies, median, mid 95th percentile range with 2.5th and 97.5th percentiles as limits. Reference intervals were estimated using nonparametric methods. No intergender statistical comparison was made.

**Funding:** The Ministry of Defence Health Implementation Programme provided support in the form of salaries for authors [OA, NO, NH, TU, AA, UE, IO, GN, IO, UA], but did not have any additional role in the study design, data collection and analysis, decision to publish, or preparation of the manuscript. The specific roles of these authors are articulated in the 'author contributions' section.

**Competing interests:** TK is affiliated with Ave Health Sense Ltd. There are no patents, products in development or marketed products to declare. This does not alter our adherence to PLOS ONE policies on sharing data and materials. There are no commercial affiliations with any organization in the funding of this study.

## Results

Complete records were obtained for 6,169 eligible participants. Median values and associated reference intervals were similar in both genders.

## Conclusion

The findings from this study will help in clinical decision-making and play a significant role in supporting the current global rapid expansion of clinical trials in response to the urgent need for preventive and therapeutic solutions to existing and emerging diseases.

## Introduction

Reference intervals are assessment tools for interpretation of clinical test results. These intervals describe the dispersion of test parameter values of apparently healthy persons in defined populations as health status indicators. These limits have been described as the most common decision support tool in laboratory medicine and the inclusion of reference intervals in a pathology report is endorsed by the international clinical laboratory standard ISO 15189, recommended by the Clinical and Laboratory Standard Institute (CSLI) and required by the College of American Pathologists (CAP), [1–4]. Though individuals could appear healthy, identification of appropriate biological markers and prevalent diseases in the population is necessary prior to determining laboratory reference intervals [3].

The use of reference intervals derived and validated in populations outside the region of application for medical decision-making and intervention could be misleading and may impact negatively on clinical management of patients or clients [1, 3]. Locally-generated laboratory reference intervals are therefore necessary for accurate diagnosis of medical disorders, disease staging, treatment monitoring, clinical trials and medical research in general. Many countries in Europe, America and Asia have established their national laboratory reference values [5–7]. However, studies from African countries are fragmented with no nationally established reference intervals. Available African reference ranges, though fragmented, have shown variation of biochemical and haematological values when compared to reference intervals validated in Western populations [8–15].

As a result of this, biochemical reference intervals commonly used in Nigeria are based on values obtained from Western-derived literature or laboratory test-kit inserts. Most available Nigerian studies on reference intervals are hospital-based or conducted on sub-regional and regional populations [8, 10, 15]. Studies of biochemical reference intervals on a truly nationally representative apparently healthy Nigerian population are unavailable. This study, which covered all the 36 States of Nigeria and the Federal Capital Territory, seeks to establish the national clinical biochemistry reference intervals among young adult Nigerians who applied for military service. It also reviews the reference intervals derived from this study with those obtained from predominantly Western cohorts in Europe and North America [16, 17].

## Materials and methods

### Study area

Nigeria is located in West Africa. The population of Nigeria is estimated to be about 206 million while the age group 18–26 years constitute about 15.9% (33 million) [18]. It is made up of 36 states and the Federal Capital Territory.

## Study design and study population

This was a cross-sectional study among a population of young Nigerian adult volunteers that applied for military service between March and October 2014. The study population comprised of adult male and female Nigerians within the age group eighteen to twenty-six years old, from all the 36 Nigerian states and the Federal Capital Territory. The total population of military applicants who gave informed consent were considered eligible for this study. However, some were excluded based on the following pre-determined exclusion criteria:

1. Presence of antibodies against the Human Immuno-deficiency virus (HIV)

2. Presence of surface antigen to Hepatitis B virus in plasma

3. Serological reactivity to Plasmodium species in plasma samples.

4. Reactivity of females to plasma B-HCG as an indication for pregnancy.

5. Presence of glycosuria, proteinuria, haematuria and bilirubinuria.

Out of 7,797 volunteers, a total of 1,628 were excluded. According to CLSI and IFCC recommendations, studies aimed at establishing Reference Intervals should have a minimum of 120 healthy participants in each category of the grouping variable [3, 13]. With the enrolment of 5932 males and 237 females, this requirement was exceeded in our study. All participants were apparently healthy young people from age 18 to 26 years from all states and the Federal Capital Territory in Nigeria.

## Specimen collection and rapid testing for biomarkers

Blood samples (14.5mls) were collected from each eligible participant for laboratory analysis, using rapid diagnostic tests to detect serological markers of infection or pregnancy. Plasma obtained from 4.5 mls of whole blood collected in EDTA bottles was used for serological testing while serum, obtained from 10 mls of blood collected in serum separator tubes (SST) was used for biochemical analysis after centrifugation. Rapid testing for HIV-1 & 2 was performed using the Nigerian national HIV serial testing algorithm (Determine®, Unigold® and Stat-Pak®); HBsAg sero-positivity and pregnancy status were determined using LabACON® kits (Citus Diagnostic Inc, British Columbia, Canada) while malaria infection screening was performed using SD BIOLINE® rapid diagnostic test kit (Standard Diagnostics Inc, Korea). Combi-9® urinalysis rapid test kit (Machery-Nagel GmbH & Co-KG, Duren, Germany) was used to detect haematuria. All assays were performed in accordance with product manufacturers' guidelines. Rapid testing for infection biomarkers was performed within 6 hours of sample collection.

## Biochemical analysis

Biochemical analysis was conducted on serum samples, extracted from 10 mls of blood, and stored at -80˚C in the Defence Reference Laboratory, Abuja, using Selectra Pro-S® automated clinical chemistry analyzer (Vital Scientific, Elitech Group Company, Netherlands). Assays performed included liver function tests (Total Serum Proteins, Albumin, Serum Globulin, alanine transaminase (ALT), aspartate transaminase (AST), Alkaline Phosphatase, gammaglutamyl transferase (GGT), Total Bilirubin, Direct and Indirect Bilirubin, Urea, Creatinine, Electrolytes (Na, K, Cl), Lipid profile (Total Cholesterols, HDL, LDL Cholesterols and Triglycerides), Calcium, Phosphate, Uric Acid, serum Lactate and serum Amylase). The auto-analyzer could not provide the results for all parameters in all the specimens (N) leading to missing

**Table 1. Analytical principles used for clinical biomarker estimation.**

| S/No | Analyte | Analytical principle |
|---|---|---|
| 1. | Sodium (Na) (mmol/L) | Dry ISE (Indirect Ion Selective Electrode) |
| 2. | Potassium (K) (mmol/L) | Dry ISE (Indirect Ion Selective Electrode) |
| 3. | Chloride (Cl) (mmol/L) | Dry ISE (Indirect Ion Selective Electrode) |
| 4. | Urea (U) (mmol/L) | Enzymatic colorimetric kinetic (Berthelot's Urease Method) |
| 5. | Creatinine (umol/L) | Colorimetric Kinetic (Jaffe's Method). |
| 6. | Lactate (mmol/L) | Colorimetric Enzymatic Endpoint (Elitech) |
| 7. | Phosphate (mmol/L) | Colorimetric Endpoint. Phosphomolybdate |
| 8. | Uric Acid (umol/L) | Colorimetric UV Kinetics (Elitech) |
| 9. | Calcium (Ca) (mmol/L) | Colorimetric Endpoint ARSENZOIII |
| 10. | Gamma GT (UL) | Colorimetric UV Kinetics IFCC-GLUPA-C |
| 11. | Amylase (U/L) | Enzymatic UV Colorimetric (Kinetic) (Elitech) |
| 12. | Total Protein (g/l) | Colorimetric End Point (Biuret's method) |
| 13. | Albumin (g/l) | Colorimetric End point, (Bromocresol Green Colorimetric Dye Binding Method). |
| 14. | Total Globulin (g/l) | Colorimetric Endpoint. Calculated with reference to Total Protein and Albumin |
| 15. | ALT (U/L) | Enzymatic UV KINETICS method. IFCC |
| 16. | AST (U/L) | Enzymatic UV KINETICS method. IFCC |
| 17. | Alkaline Phosphatase (U/L) | Enzymatic UV Kinetic method IFCC |
| 18. | Total Bilirubin (umol/L) | Colorimetric End- Point (Modified Malloy and Evelyn) |
| 19. | Direct Bilirubin (umol/L) | Colorimetric End- Point. (Modified Malloy and Evelyn) Sulfanilic acid method |
| 20. | Indirect Bilirubin (umol/L) | Colorimetric End- Point. (Modified Malloy and Evelyn) Sulfanilic acid method |
| 21. | Total Cholesterol (mmol/L) | Enzymatic, colorimetric End point (CHOD-PAP) |
| 22. | HDL Cholesterol (mmol/L) | Direct Selective Detergent (Elitech) |
| 23. | LDL Cholesterol (mmol/L) | Direct Selective Detergent (Elitech) |
| 24. | Triglycerides (mmol/L) | Enzymatic, End point (Elitech) |

values for some parameters. The analytical principles utilized for estimating the electrolytes are as shown in Table 1.

## Quality control

Laboratorians who had consistently passed Competency Testing for at least 3 years prior to study commencement were drawn from the Defence Reference Laboratory (DRL), Abuja, to carry out initial biomarker screening in the field and subsequent biochemical analysis in the DRL. Routine Quality Control (QC) procedures such as equipment/assay validation and use of commercially prepared controls were strictly adhered to. Data confidentiality and quality were ensured through specimen de-identification/aggregation and double entry by 2 independent Data Entry Clerks.

## Statistical analysis

Data sets were entered into MS Excel® and exported to SPSS® for analysis. Frequencies and proportions were generated and reference intervals were computed independently for male and female participants. Non-parametric method of analysis was used to generate the measure

of central tendency (median) and dispersion (2.5th– 97.5th range) for each biochemical parameter. The results were stratified by gender for each parameter, no statistical comparison was made and therefore no statistical test of significance was required.

## Ethical considerations

Ethical approval was obtained from the Nigerian Ministry of Defense Health Research Ethics Committee (MODHREC) Abuja. Written informed consent was obtained from each volunteer followed by HIV pre- and post-test counseling. Participants who tested positive to bio-markers of infection were referred to the nearest care and treatment centre. Personal identifiers were eliminated through the use of unique Study Identification Numbers (SIN).

## Results

A total of 6,169 (5,932 males and 237 females) participants were enrolled in this study.

The gender-differentiated and aggregate sample size, median, and 95 percentile reference ranges for selected general biochemical parameters are shown in Table 2. All median values and associated reference ranges are shown for males only, females only and a total figure for both genders. Table 3 shows similar findings for selected liver function and lipid parameters.

The findings in this study are shown against standard Western reference intervals in Tables 4 and 5. Only the 95 percentile interval for combined male and female parameters are listed in Tables 4 and 5 since the comparative Western values were not segregated by gender, except for serum Creatinine, Uric Acid, Gamma GT, Alanine and Aspartate Transaminases. The comparative western values for Total Globulin, Direct Bilirubin and Indirect Bilirubin were not contained in the cited British and American publications.

## Discussion

This study set out to determine the biochemical reference ranges among young adult Nigerians.

The reference intervals from this study appear generally wider than the western ranges, with consistently lower limits for most parameters. The upper limits for a number of parameters in our study appeared notably higher than the western reference values especially for Potassium, Creatinine, Lactate, Phosphate and Alkaline Phosphatase published in standard works [16, 17, 19]. Although median values and associated reference ranges are apparently similar in most male and female parameters, there appears to be a marked difference between the median values for alkaline phosphatase in both genders even though both reference ranges are in full overlap. The 95 percentile reference limits for Alkaline Phosphatase in our study are much higher than those of comparative Western populations. The range between lower and upper limits is also considerably wider with the finding of 259 against 66 and 100 U/L for American and British populations respectively in this study as reflected in Table 4.

Our findings for all other parameters are generally similar to Western reference intervals [16, 17, 19] except for total globulin and bilirubin (direct and indirect), whose comparative western values were unavailable in the cited studies. The upper reference limit of 7.0 mmol/L for potassium is particularly notable in the young adult participants of this study. This finding cannot be easily attributed to changes in renal excretion due to older age or hormones that regulate its clearance. While the reasons for this are not very clear, similar work done by Miri-Dashe's group on normal Nigerian adults [8] showed potassium reference ranges of 4.0–7.5 and 4.0–7.7 mmol/L for males and females respectively.

The high reference intervals for creatinine observed in this study may be due to dietary factors and high degree of physical activity that is common among the young adult age group of

**Table 2. Statistical analysis of selected general biochemical reference intervals among young adult Nigerians.**

| Analytes | Gender | N | Median | Reference Value (95% Percentile) |
|---|---|---|---|---|
| **Sodium (Na) mmol/L** | Male (M) | 5932 | 133 | 114–144 |
| | Female (F) | 237 | 133 | 114–146 |
| | M and F | 6169 | 133 | 114–144 |
| **Potassium (K) mmol/L** | Male | 5932 | 4.0 | 2.7–7.0 |
| | Female | 237 | 4.1 | 2.8–7.0 |
| | M and F | 6169 | 4.0 | 2.7–7.0 |
| **Chloride (Cl) mmol/L** | Male | 5929 | 93 | 80–100 |
| | Female | 237 | 93 | 79–100 |
| | M and F | 6166 | 93 | 80–100 |
| **Urea (U) mmol/L** | Male | 5931 | 3.6 | 1.2–9.0 |
| | Female | 237 | 3.4 | 1.3–7.2 |
| | M and F | 6168 | 3.6 | 1.2–9.0 |
| **Creatinine (Cr) umol/L** | Male | 5932 | 88 | 46–132 |
| | Female | 237 | 90 | 48–136 |
| | M and F | 6169 | 88 | 46–132 |
| **Lactate (L) mmol/L** | Male | 5926 | 3.2 | 1.4–8.2 |
| | Female | 237 | 3.2 | 1.0–7.6 |
| | M and F | 6163 | 3.2 | 1.4–8.1 |
| **Phosphate (P) mmol/L** | Male | 5930 | 1.3 | 0.6–4.8 |
| | Female | 237 | 1.3 | 0.6–4.6 |
| | M and F | 6167 | 1.3 | 0.6–4.7 |
| **Uric Acid (UA) umol/L** | Male | 5931 | 274 | 141–462 |
| | Female | 237 | 273 | 140–400 |
| | M and F | 6168 | 274 | 141–460 |
| **Calcium (Ca) mmol/L** | Male | 5932 | 2.3 | 1.6–2.7 |
| | Female | 237 | 2.3 | 1.6–2.6 |
| | M and F | 6169 | 2.3 | 1.6–2.7 |
| **Gamma GT (GGT) U/L** | Male | 5899 | 22 | 10–55 |
| | Female | 236 | 23 | 11–52 |
| | M and F | 6135 | 22 | 10–55 |
| **Amylase (AMYL) U/L** | Male | 5924 | 58 | 24–117 |
| | Female | 237 | 56 | 24–116 |
| | M and F | 6161 | 58 | 24–117 |

the participants in this study. The high values obtained for phosphate, lactate and alkaline phosphatase are also consistent with the expected pattern for the young adult age group enrolled in this study. These analytes are related to bone metabolism which is very robust at the age range of the study population [20]. Enzymatic activity levels for alkaline phosphatase, calcium and phosphate have been reported to achieve their highest peak during main bone growth period before a slow decline sets in from puberty [20]. These findings from our study are in keeping with such expected trends.

Nigerian public health laboratories, like most clinical laboratories in Africa, rely on reference intervals obtained from textbooks, instrument manuals and reagent inserts for interpretation of laboratory results [8, 9, 13, 14]. This study has shown decreased lower reference limits and higher upper limits with wider overall reference intervals for most of our study analytes when compared to western values. Since Reference Intervals are intended to inform the clinician that laboratory values within the interval indicate a non-diseased condition [21], the

**Table 3. Statistical analysis of selected liver function and lipid parameters among young adult Nigerians.**

| Analytes | Gender | N | Median | Reference Value (95% Percentile) |
|---|---|---|---|---|
| **Total Protein (TP) g/L** | Male | 5930 | 79.5 | 52.2–98.0 |
| | Female | 237 | 79.3 | 54.5–97.4 |
| | M and F | 6167 | 79.5 | 52.3–97.9 |
| **Albumin (ALB) g/L** | Male | 5898 | 45.8 | 34.8–53.0 |
| | Female | 237 | 45.9 | 33.9–54.2 |
| | M and F | 6135 | 45.8 | 34.8–53.0 |
| **Globulin (GLOB) g/L** | Male | 5929 | 33.0 | 11.6–52.3 |
| | Female | 237 | 33.8 | 10.3–49.0 |
| | M and F | 6166 | 33.0 | 11.5–51.9 |
| **ALT (U/L)** | Male | 5928 | 16.0 | 3.5–63.5 |
| | Female | 237 | 16.6 | 4.5–63.0 |
| | M and F | 6165 | 16.0 | 3.5–63.2 |
| **AST (U/L)** | Male | 5931 | 18.3 | 2.9–52.0 |
| | Female | 237 | 19.0 | 3.2–47.4 |
| | M and F | 6168 | 18.3 | 2.9–51.4 |
| **Alkaline Phos (ALP) U/L** | Male | 5926 | 162 | 79–338 |
| | Female | 237 | 152 | 86–327 |
| | M and F | 6163 | 162 | 79–338 |
| **Total Bilirubin (TBIL) umol/L** | Male | 5932 | 8.8 | 1.5–33.8 |
| | Female | 237 | 8.9 | 1.4–41.2 |
| | M and F | 6169 | 8.8 | 1.5–33.8 |
| **Direct Bilirubin (DBil) umol/L** | Male | 5932 | 4.3 | 0.4–16.9 |
| | Female | 237 | 4.3 | 0.2–19.6 |
| | M and F | 6169 | 4.3 | 0.4–17.0 |
| **Indirect Bilirubin (IBILI) umol/L** | Male | 5922 | 4.0 | 0.4–16.6 |
| | Female | 237 | 4.3 | 0.4–15.4 |
| | M and F | 6159 | 4.0 | 0.4–16.5 |
| **Total Chol (TChol) mmol/L** | Male | 5928 | 4.0 | 2.4–6.2 |
| | Female | 237 | 4.0 | 2.2–6.3 |
| | M and F | 6165 | 4.0 | 2.4–6.2 |
| **HDL Chol (mmol/L)** | Male | 5932 | 1.3 | 0.5–2.7 |
| | Female | 237 | 1.2 | 0.4–2.3 |
| | M and F | 6169 | 1.3 | 0.5–2.7 |
| **LDL Chol (mmol/L)** | Male | 5925 | 2.3 | 0.8–4.5 |
| | Female | 236 | 2.4 | 0.9–4.7 |
| | M and F | 6161 | 2.3 | 0.8–4.5 |
| **Triglycerides (TG) mmol/L** | Male | 5930 | 0.8 | 0.5–1.8 |
| | Female | 237 | 0.7 | 0.4–1.6 |
| | M and F | 6167 | 0.8 | 0.5–1.8 |

significance of this finding lies in the fact that many young adult Nigerians who would otherwise have been classified as having pathological laboratory results can now have better interpretation of their clinical biochemical tests.

This will also impact positively on the eligibility of study volunteers willing to participate in preventive and therapeutic clinical trials as well as their monitoring for adverse reaction especially in the contemporary era of widespread research into existing and emerging infectious diseases. The results of this study will also serve as benchmark reference intervals for analytes

**Table 4. Common clinical biochemistry reference intervals of young Nigerian adults and standard reference values.**

| Parameters | | This Study | Standard Reference Intervals | | |
| --- | --- | --- | --- | --- | --- |
| | | | Dosso [14] (Ghana) | Tietz [16] (American) | NHS [17] (British) |
| Sodium (Na) (mmol/L) | | 114–144 | 135–150 | 137–143 | 133–146 |
| Potassium (K) (mmol/L) | | 2.7–7.0 | 3.6–5.2 | 3.8–4.9 | 3.5–5.3 |
| Chloride (Cl) (mmol/L) | | 80–100 | 102–114 | 101–106 | 95–108 |
| Urea (U) (mmol/L) | | 1.2–9.0 | 0.9–5.7 | 3.3–7.9 | 2.5–6.5 |
| Creatinine (umol/L): | Male | 46–132 | 56–119 | 80–115 | 59–104 |
| | Female | 48–136 | 47–110 | 53–97 | 45–84 |
| Lactate (mmol/L) | | 1.4–8.1 | N/A | 1.78–1.88 | 0.5–2.2 |
| Phosphate (mmol/L) | | 0.6–4.8 | 0.7–1.5 | 0.95–1.52 | 0.8–1.5 |
| Uric Acid (umol/L): | Male | 141–462 | 126–418 | 218–459 | 200–430 |
| | Female | 140–400 | 83–381 | 147–366 | 140–360 |
| Calcium (Ca) (mmol/L) | | 1.60–2.70 | N/A | 2.28–2.60 | 2.20–2.60 |
| Gamma GT (UL): | Male | 10–55 | 9–71 | 12–62 | 0–60 |
| | Female | 11–52 | 6–53 | 12–38 | 0–40 |
| Amylase (U/L) | | 24–117 | 32–139 | 31–107 | 28–100 |

Comparative Reference Interval sources: References [14, 16, 17].

such as globulin fraction, direct and indirect bilirubin which are not readily available for African populations.

## Limitations

This study was conducted with volunteers for military service in Nigeria. Due to the rigors of military training and service, these volunteers are expected to be at peak physical condition

**Table 5. Common biochemical reference intervals for liver function, lipid parameters and standard reference values.**

| Parameters | | This Study | Standard Reference Intervals | | |
| --- | --- | --- | --- | --- | --- |
| | | | Dosso [14] (Ghana) | Tietz [16] (American) | NHS [17] (British) |
| Total Protein (g/l) | | 52–98 | 50.6–86.7 | 65–83 | N/A |
| Albumin (g/l) | | 35–53 | 33.0–49.9 | 46–53 | 35–50 |
| Total Globulin (g/l) | | Dec-52 | | N/A | N/A |
| ALT (U/L) | Male | 4–63 | 8–54 | 18–78 | 10–50 |
| | Female | 5–63 | 6–51 | 14–41 | 10–35 |
| AST (U/L) | Male | Mar-52 | 17–60 | 18–54 | 0–40 |
| | Female | Mar-47 | 13–48 | 18–34 | 0–32 |
| Alkaline Phosphatase (U/L) | | 79–338 | 85–241 | 50–116 | 30–130 |
| Total Bilirubin (umol/L) | | Feb-34 | 2.9–25.8 | 3–18 | 0–21 |
| Direct Bilirubin (umol/L) | | 0–17 | 0.8–4.0 | N/A | N/A |
| Indirect Bilirubin (umol/L) | | 0–17 | N/A | N/A | N/A |
| Total Cholesterol (mmol/L) | | 2.4–6.2 | 2.0–5.4 | 3.0–5.9 | 0–5.0 |
| HDL Cholestrol (mmol/L) | | 0.5–2.7 | N/A | 0.8–1.8 | 1.2–3.0 |
| LDL Cholesterol (mmol/L) | | 0.8–4.5 | N/A | 1.6–4.9 | 1.0–3.0 |
| Triglycerides (mmol/L) | | 0.5–1.8 | N/A | 0.4–2.1 | 0–1.7 |

Comparative Reference Interval sources: References [14, 16, 17].

and optimal health. Therefore, they are likely to be fitter and 'more' healthy than other apparently healthy individuals in the general population thus giving rise to a healthy volunteer bias. We however do not consider this to have significantly altered our findings as the values of biochemical parameters in the body are maintained within reasonably defined limits in the absence of significant pathology.

Samples used for this study could not be analyzed immediately after collection due to huge sample size, large number of analytes to be tested and the big geographical size of Nigeria. Samples were stored at -80˚C until the time for analysis to maintain specimen integrity. Body mass index (BMI) and other possible confounding factors like smoking habit, alcohol consumption and dietary pattern [9, 14] could not be obtained. Wider age stratification to include children and the elderly group was also not possible limiting the generalizability of findings to these age groups.

## Conclusion

This study has met the objective of establishing reference intervals for common biochemistry parameters among young adult Nigerians. This will aid proper clinical decision-making process and play a significant role in supporting the current global rapid expansion of clinical trials in response to the urgent need for preventive and therapeutic solutions to existing and emerging diseases where locally generated reference ranges are of paramount importance. We advocate that similar studies be conducted regularly and on a wider age stratification of participants across the country in an attempt to build a culture of more reliance on locally derived laboratory reference ranges. Confidence in and acceptance of locally generated reference intervals will be facilitated by post-study validation and comparison of flagging rates of the relevant laboratory systems.

## Supporting information

**S1 File. Author list and contributions.**
(DOCX)

**S2 File. Biochemistry dataset.**
(SAV)

**S3 File. MODHREC approval—Jan 2014.**
(PDF)

**S1 Appendix. Consent form.**
(DOCX)

## Author Contributions

**Conceptualization:** Ojor Ayemoba, Nurudeen Hussain, Tahir Umar, Anthony Ajemba-Life, Terfa Kene.

**Data curation:** Terfa Kene, Usman Adekanye.

**Formal analysis:** Terfa Kene, Uchechukwu Edom, Ikechukwu Ogueri, Goodluck Nwagbara, Inalegwu Ochai, Ikenna Onoh.

**Investigation:** Uchechukwu Edom, Ikechukwu Ogueri, Goodluck Nwagbara, Inalegwu Ochai.

**Methodology:** Ojor Ayemoba.

**Project administration:** Ojor Ayemoba, Nathan Okeji, Nurudeen Hussain, Tahir Umar, Anthony Ajemba-Life.

**Supervision:** Ojor Ayemoba, Nathan Okeji.

**Validation:** Ojor Ayemoba, Ikechukwu Ogueri, Usman Adekanye, Ikenna Onoh.

**Writing – original draft:** Ojor Ayemoba, Terfa Kene.

**Writing – review & editing:** Ojor Ayemoba, Nathan Okeji, Nurudeen Hussain, Tahir Umar, Anthony Ajemba-Life, Terfa Kene, Uchechukwu Edom, Ikechukwu Ogueri, Goodluck Nwagbara, Inalegwu Ochai, Usman Adekanye, Ikenna Onoh.

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
