## [Decision Letter · Decision Letter 0]

13 Oct 2020

PONE-D-20-25237

Reference intervals of common clinical biochemistry analytes in young Nigerian adults

PLOS ONE

Dear Dr. Ayemoba,

Thank you for submitting your manuscript to PLOS ONE. After careful consideration, we feel that it has merit but does not fully meet PLOS ONE’s publication criteria as it currently stands. Therefore, we invite you to submit a revised version of the manuscript that addresses the points raised during the review process.

Both reviewers have identified multiple problems and missing information in the Methods section of your manuscript. Additionally, the statistical analyses need to be reviewed. Be sure to have your manuscript edited by a native English speaking scientist.

We look forward to receiving your revised manuscript.

Kind regards,

Ulrike Gertrud Munderloh, Ph.D.

Academic Editor

PLOS ONE

Journal Requirements:

'This study was funded by the Nigerian Ministry of Defence Health Implementation Programme, No 4B Ikole Street, Area 11, Abuja, FCT, Nigeria. The funder had no role in study design, data collection and analysis, decision to publish, or preparation of the manuscript'

We note that one or more of the authors are employed by a commercial company: Ave Health Sense Ltd.

Reviewers' comments:

Reviewer's Responses to Questions

**Comments to the Author**

1. Is the manuscript technically sound, and do the data support the conclusions?

Reviewer #1: No

Reviewer #2: Partly

2. Has the statistical analysis been performed appropriately and rigorously? 

Reviewer #1: No

Reviewer #2: No

3. Have the authors made all data underlying the findings in their manuscript fully available?

Reviewer #1: Yes

Reviewer #2: Yes

4. Is the manuscript presented in an intelligible fashion and written in standard English?

Reviewer #1: No

Reviewer #2: No

5. Review Comments to the Author

Reviewer #1: 1. Abstract, methods: You stated the performed tests was “to eliminate possible confounding conditions”, but I think it was rather to select apparently healthy individuals. In addition, this section did not include other major components of methods and materials section like study period, sampling technique, statistical tools, etc.

2. Abstract, results: The third statement is not relevant, better to remove it.

3. Sample size: I think your sample size was 6,169 not 8,505.

4. Introduction (1st paragraph): Please combine the separately written idea as 1, 2, 3, 4. The message is similar.

5. Introduction (paragraph 2, line 6). Are you confident to conclude as many African countries have established their own RI? Please justify your conclusion through evidence. There are some fragmented studies but it is difficult to conclude.

6. Introduction: This portion did not clearly show the gap (such as factors affecting RI not included). Is national RI recommended in such multiethnic country like Nigeria?

7. Materials and methods: You have to follow scientifically accepted guidelines in order to recruit apparently healthy participants. There is no sampling technique mentioned in the document. How did you check the data distribution and partition?

8. Result: The first paragraph should be removed and incorporated in materials and methods section.

9. Result, 2nd paragraph, last statement and 3rd paragraph, 4th and 5ith statement: Interpretation of data is not recommended. Data should be interpreted in the discussion section.

10. Result: I think the RI of females is dominated by males due to large sample size. How could you justify this?

11. Result: Table 3&4: You try to compare your result with American and British studies only. You have to include studies in Africa, Asia, Latin, Etc. Besides, the American study is a text book study which is not appropriate. What was your base (statistics) for comparison? it is not included in materials and methods section.

12. Discussion: Which statistics you computed indicated to say ‘wider’, ‘lower’, ‘higher’ and ‘in concordance’? The discussion section is not supported by statistics and evidences done worldwide.

13. Conclusion: It is out of your result.

Reviewer #2: I congratulate the authors for established RIs for Nigerian Adults and have country specific RIs. However, I do have a few comments for the authors which I believe will improve the manuscript.

Abstract:

Reference intervals are not quantitative data. These are assessment tool to interpret test results.

Please mention the number of analytes for which reference intervals were established and age of the participants clearly in the objective instead of "young Nigerian Adults".

In the method, the authors mentioned that the samples were drawn form 36 state and Territory. However, in the methods section of the paper, the authors mentioned that these participants were volunteers who applied for military services. So the samples were not drawn? Please be consistent.

Please revise the sentence "RIs of common clinical biochemistry analytes were computed." to "RIS were estimated using nonparametric methods."

Results. "No statistical comparison was made" should be in the methods. This is not a result.

Introduction:

This is now well established that RIs are not an indicator of good health or disease unlike clinical decision limit. RIs simply indicate that certain percentage of people within a range. The comment made in the abstract about quantitative data also apply. Please revise the 1st sentence.

How is the volunteers who applied for military service representative of the 18 - 26 year population in Nigeria?

what is the justification of comparing the RIs with western cohorts in Europe and North America but not the RIs currently used in Nigeria?

Methods:

The description of Nigeria may not be necessary as part of the Study Area. Instead of total population I would report the 18 - 26 years old population to who these RIs are applicable.

I don't think the study was design prospectively and desired number of sample was determined. In fact, the study used the data available from the screening process. So the sample size determination may not be appropriate. I would advise the authors to move this point to the discussion expressing that you have more sample than what is recommended.

Biochemical Analysis: Please mention the analytical principle used for each of these biomarkers either as part of the paper or as supplementary. This information will be use for the laboratories who wish to use the RIs knowing there is variation in reference values between analysers/instruments.

Statistical analysis:

Have the authors excluded any outliers? If yes, please explain what method was applied. If not, please justify who not. This is an important step of the statistical analysis.

The authors should clearly mention what method was used for analysis. Measures of Central Tendency include Mean, Median, Mode etc. and deviation can be range, SD, Variance etc. I understand the authors have used median and 2.5th and 97.5th centile which is defined as nonparametric method by CLSI. So this should be clearly mentioned.

Please report the 95% CIs of the RIs. This has also been recommend in recent publications.

What method was applied to compare the established RIs with western countries? Visual? by considering with 95% CI of the established RIs include western countries' RIs?

Results:

Why the auto-analyser could not provide results for all parameters?

Considering the biochemistry analytes are associated with age in the paediatric population i.e. age 0 - <18, is there any association with age? The authors should include scatter plot of analytes across age by sex in the supplementary for the readers to understand age-specific RIs were not necessary for the young adult.

One of the concern I have is the unequal distribution of sexes in the sample. Overall only 237 samples were female. Hence, the RIs of male will be more precise compared to female.

Discussion & Conclusion: While the authors advocate for similar studies, validation or comparison of flagging rate would be another important step in ensuring the RIs are adopted by local laboratories.

6. PLOS authors have the option to publish the peer review history of their article (what does this mean?). If published, this will include your full peer review and any attached files.

Reviewer #1: No

Reviewer #2: No

---

## [Author Response · Author response to Decision Letter 0]

21 Jan 2021

19th January 2021

Rebuttal Letter to PLOS ONE

Response to Reviewers

Dear Ulrike Gertrud Munderloh,

Thank you for the careful review of our manuscript. After careful consideration of the reviews, I hereby write this rebuttal letter to respond to the points and comments raised by the reviewers. A revised version of the manuscript and the track changes are attached as annexes I and II. 

We would like to bring to your attention a typographical error with respect to the number of volunteers who were initially considered for the study. The correct figure is 7,797 as opposed to 8,505 in earlier versions of the manuscript. This has been corrected in the submitted manuscript and did not affect the final number of volunteers eventually enrolled (6,169) nor the final results presented.

The following are the responses to the points raised by the reviewers:

We note that one or more of the authors are employed by a commercial company: Ave Health Sense Ltd.

Response: At the point of conceptualization and implementation of the field activities, the author was a staff of US Department of Defence Walter Reed Programme in Nigeria, which is one of our affiliate organizations.

The author works part-time with Ave Health Sense Ltd which is his current fixed address for delivery of physical mails.

He contributed in the past to protocol development and field data collection. He more recently contributed to data extraction, analysis and writing of the manuscript in his personal capacity.

Ave Health Sense Ltd has NOT contributed funding for the research in whatever including but not limited to payment of salaries. 

Also, Dr Ikenna Onoh, who is one of the authors, is a Nigerian Filed Epidemiology Laboratory Training Programme (NFELTP) resident attached to Ministry of Defence Health Implementation Programme as part of his field training. There are no financial obligations to his participation in this study. 

Response: There are no commercial affiliations with any organization in the funding of this study.

Response: “The Ministry of Defence Health Implementation Programme provided support in the form of salaries for authors [OA, NO, NH, TU, AA, UE, IO, GN, IO, UA], but did not have any additional role in the study design, data collection and analysis, decision to publish, or preparation of the manuscript. The specific roles of these authors are articulated in the ‘author contributions’ section.”

 Response: “There are no commercial affiliations in the funding of this study”

Please also provide an updated Competing Interests Statement declaring this commercial affiliation along with any other relevant declarations relating to employment, consultancy, patents, products in development, or marketed products, etc. 

 Response: "TK is affiliated with Ave Health Sense Ltd. There are no patents, products in development or marketed products to declare. This does not alter our adherence to PLOS ONE policies on sharing data and materials. There are no commercial affiliations with any organization in the funding of this study."

 Response: "This has been captured in the updated Competing Interests Statement above.”

Response: The updated funding statement is as indicated above. The updated Competing Interests Statement is as shown above.

Reviewers' comments:

Reviewer's Responses to Questions

Comments to the Author

Review Comments to the Author

Reviewer #1: 

1. Abstract, methods: You stated the performed tests was “to eliminate possible confounding conditions”, but I think it was rather to select apparently healthy individuals. In addition, this section did not include other major components of methods and materials section like study period, sampling technique, statistical tools, etc.

Responses – 

- “to eliminate possible confounding conditions” has been replaced with “restrict our study population to apparently healthy participants.”.

- Study period - The study period for this study is included in the method section of the manuscript (This was a cross-sectional study in which data was collected between March and October 2014, laboratory and data analyses were completed in 2018)

- Sampling technique – The sampling technique has been added to the abstract and now reads “It was a total study among volunteers for military service”

- Statistical tools – The statistical tools used have been added to the abstract (frequencies, median, mid 95th percentile range with 2.5th and 97.5th percentiles as limits)

2. Abstract, results: The third statement is not relevant, better to remove it.

Response – The third statement in abstract result has been deleted

3. Sample size: I think your sample size was 6,169 not 8,505.

Response - The number of participants that consented to participate in this study were 7,797 (note correction above). However, only 6,169 were eligible following serological and other tests to eliminate those with prevalent health conditions in Nigeria, which are capable of affecting health indices (as stated in the 3rd sentence of the Methods section of the Abstract).

4. Introduction (1st paragraph): Please combine the separately written idea as 1, 2, 3, 4. The message is similar.

Response¬ – Correction effected

5. Introduction (paragraph 2, line 6). Are you confident to conclude as many African countries have established their own RI? Please justify your conclusion through evidence. There are some fragmented studies but it is difficult to conclude.

Response – The sentence has been modified and now reads – ”However, studies from African countries are fragmented with no nationally established reference intervals….(8-15)” 

6. Introduction: This portion did not clearly show the gap (such as factors affecting RI not included). Is national RI recommended in such multiethnic country like Nigeria?

Response – Nigeria has diverse ethnic groups with different cultural and social practices including varying economic classes. All these may affect dietary and developmental practices which can alter RIs. However, in accordance with studies from Western countries (such as the UK and the US), single RI values are used despite the fact that different races exist. Nigeria has a diverse social, cultural, environmental, dietary and economic practices which may affect RI in addition to prevalent health conditions in the country. Apart from the health conditions, these other variables that may affect RI could not be explored. 

7. Materials and methods: You have to follow scientifically accepted guidelines in order to recruit apparently healthy participants. There is no sampling technique mentioned in the document. How did you check the data distribution and partition?

Response – This was a total population study among applicants for military service. No probability sampling was done in the wider source population nor from the participants as all of them were considered eligible subject to granting informed consent. Some of the participants that had bio-markers for known pathologies/conditions were excluded. We used a non-parametric analytic method to generate estimates and associated intervals. These methods do not need an assumption of normal distribution. We did not partition the data and we used a common reference interval for each analyte in both males and females as there was no statistically significant difference in the RIs.

Two new sentences were included to capture the idea more correctly. The inclusion criteria was also changed to exclusion criteria.

8. Result: The first paragraph should be removed and incorporated in materials and methods section.

Response – 

The first paragraph in the results section has been incorporated into methods section. An introductory sentence on the final number of participants studied was added to the results. 

9. Result, 2nd paragraph, last statement and 3rd paragraph, 4th and 5ith statement: Interpretation of data is not recommended. Data should be interpreted in the discussion section.

Response – This observation is noted and the statements have been moved to the discussion section as suggested.

10. Result: I think the RI of females is dominated by males due to large sample size. How could you justify this?

Response – The recommended CLSI minimum sample size standards for RI is 120 individuals per sub-group of a grouping variable. This study population exceeded this required minimum. This study was non-interventional and statistical comparison by gender was not done. Therefore, balance between the two groups was not necessary. 

11. Result: Table 3&4: You try to compare your result with American and British studies only. You have to include studies in Africa, Asia, Latin, Etc. Besides, the American study is a text book study which is not appropriate. What was your base (statistics) for comparison? it is not included in materials and methods section.

Response - In Nigeria, due to lack of locally generated national RIs, like in most African countries, our clinical result interpretations are based on Western figures mainly represented by British and American values.. This was the reason for comparison with Western figures and why we did not consider it necessary to compare with Asian and Latin American values. However, we have included RIs from Ghana, West Africa. There was no statistical comparison between the other countries’ RIs and this study as the primary data set for the countries were not available to us. The American RIs from the Tietz textbook are a compilation of values from individual studies published in peer-reviewed journals. No attempt was made to statistically compare our findings with these textbook values. We only eyeballed our study’s RIs with those that are currently in clinical use in our environment. This was to ascertain the suitability of these foreign values for our clinical use.

12. Discussion: Which statistics you computed indicated to say ‘wider’, ‘lower’, ‘higher’ and ‘in concordance’? The discussion section is not supported by statistics and evidences done worldwide.

Response: The second sentence of the discussion states that our values APPEAR to be “wider”, “lower” and “higher” when juxtaposed with western values. By this, we do not imply statistically assessed relationships. This study, like has been mentioned was purely descriptive and involved only eyeballing.

13. Conclusion: It is out of your result.

Response: The conclusion has been adjusted to reflect the reviewer’s comments as advised.

Reviewer #2: I congratulate the authors for established RIs for Nigerian Adults and have country specific RIs. However, I do have a few comments for the authors which I believe will improve the manuscript.

Abstract:

Reference intervals are not quantitative data. These are assessment tool to interpret test results.

Response: Correction has been effected and the sentence has been modified

Please mention the number of analytes for which reference intervals were established and age of the participants clearly in the objective instead of "young Nigerian Adults".

Response: This has been addressed and the number of analytes (24) have been included as well as the age group.

In the method, the authors mentioned that the samples were drawn form 36 state and Territory. However, in the methods section of the paper, the authors mentioned that these participants were volunteers who applied for military services. So the samples were not drawn? Please be consistent.

Response: 

The word “drawn” has been deleted and a new sentence “It was a total study among volunteers for military service” has been added. 

Please revise the sentence "RIs of common clinical biochemistry analytes were computed." to "RIS were estimated using nonparametric methods."

Response: Correction effected.

Results. "No statistical comparison was made" should be in the methods. This is not a result.

Response: The sentence has been moved to methods

Introduction:

This is now well established that RIs are not an indicator of good health or disease unlike clinical decision limit. RIs simply indicate that certain percentage of people within a range. The comment made in the abstract about quantitative data also apply. Please revise the 1st sentence.

Response: The correction has been effected

How is the volunteers who applied for military service representative of the 18 - 26 year population in Nigeria?

Response: Studies conducted using volunteers are liable to experience the ‘healthy volunteer effect/bias’. This would have played some role in this study as military applicants are expected to be generally fitter and ‘more’ healthy. However, we do not perceive this to significantly alter our findings. We have also acknowledged this limitation in the discussion section of the paper.

What is the justification of comparing the RIs with western cohorts in Europe and North America but not the RIs currently used in Nigeria?

Response: In Nigeria, due to lack of locally generated national RIs, like in most African countries, our clinical result interpretations are based on Western figures mainly represented by British and American values. This is also mentioned in the first sentence of the last paragraph of the introduction.

Methods:

The description of Nigeria may not be necessary as part of the Study Area. Instead of total population I would report the 18 - 26 years old population to who these RIs are applicable.

Response: 

The study area has been modified to reflect the population of interest in this study. We also deleted information about Nigeria that may not be necessary as advised.

I don't think the study was design prospectively and desired number of sample was determined. In fact, the study used the data available from the screening process. So the sample size determination may not be appropriate. I would advise the authors to move this point to the discussion expressing that you have more sample than what is recommended.

Response: The sub-heading ‘Sample size determination’ has been deleted and the content of that sub-section has been merged with the study design and study population section. 

Biochemical Analysis: Please mention the analytical principle used for each of these biomarkers either as part of the paper or as supplementary. This information will be use for the laboratories who wish to use the RIs knowing there is variation in reference values between analysers/instruments.

Response: A table indicating the analytical principles for each biomarker has been added to the biochemical analysis section under Methods.

Statistical analysis:

Have the authors excluded any outliers? If yes, please explain what method was applied. If not, please justify who not. This is an important step of the statistical analysis.

Response: We did not assess the data for outliers, based on the fact that the participants had been screened earlier for prevalent health conditions that may account for out-of-range values. The data was carefully cleaned and no extreme values were observed during the process.

The authors should clearly mention what method was used for analysis. Measures of Central Tendency include Mean, Median, Mode etc. and deviation can be range, SD, Variance etc. I understand the authors have used median and 2.5th and 97.5th centile which is defined as nonparametric method by CLSI. So this should be clearly mentioned.

Response: This section has been modified and the 3rd sentence in the section now reads – “Non-parametric method of analysis was used to generate the measure of central tendency (median) and dispersion (2.5th – 97.5th range) for each biochemical parameter.”

Please report the 95% CIs of the RIs. This has also been recommend in recent publications.

Response: The calculation of CI around reference limits is based on an assumption of random sampling and this was not fulfilled in this study.

What method was applied to compare the established RIs with western countries? Visual? by considering with 95% CI of the established RIs include western countries' RIs?

Response: The comparison with Western values was visual and did not involve any analytical statistics

Results:

Why the auto-analyser could not provide results for all parameters?

Response: The auto-analyzer could not provide the results for all parameters in all the specimens despite repeated runs. We did not use manual techniques to prevent inconsistencies with the study protocol.

Considering the biochemistry analytes are associated with age in the paediatric population i.e. age 0 - <18, is there any association with age? The authors should include scatter plot of analytes across age by sex in the supplementary for the readers to understand age-specific RIs were not necessary for the young adult.

Response: It has been variously established over time that among young adults of the age range in our study, values of the assessed analytes do not vary with age. For this reason, we have not generated age-specific RIs. We therefore do not consider presentation of scatter plots necessary.

One of the concern I have is the unequal distribution of sexes in the sample. Overall only 237 samples were female. Hence, the RIs of male will be more precise compared to female.

Response: The recommended CLSI minimum sample size standards for RI is 120 individuals per sub-group of a grouping variable. The RIs generated for males will be more precise than the females even though the female values meet the minimum requirement for precision. This does not affect the validity of the generated RIs in females for clinical use.

Discussion & Conclusion: While the authors advocate for similar studies, validation or comparison of flagging rate would be another important step in ensuring the RIs are adopted by local laboratories.

Response: The Defence Reference Lab where the sample analysis was conducted is internationally accredited by A2LA. This means equipment, reagents, personnel proficiency, external quality assessments and other QA procedures are carried out as necessary. A new concluding sentence has been added to the conclusion to capture this suggestion. 

Thank you very much for your time and I look forward to a feedback after review.

 OR AYEMOBA

 Brig Gen (rtd)

 Corresponding Author

---

## [Decision Letter · Decision Letter 1]

11 Feb 2021

Reference intervals of common clinical biochemistry analytes in young Nigerian adults

PONE-D-20-25237R1

Dear Dr. Ayemoba,

We’re pleased to inform you that your manuscript has been judged scientifically suitable for publication and will be formally accepted for publication once it meets all outstanding technical requirements.

Kind regards,

Ulrike Gertrud Munderloh, Ph.D.

Academic Editor

PLOS ONE

Additional Editor Comments (optional):

Reviewers' comments:

Reviewer's Responses to Questions

**Comments to the Author**

1. If the authors have adequately addressed your comments raised in a previous round of review and you feel that this manuscript is now acceptable for publication, you may indicate that here to bypass the “Comments to the Author” section, enter your conflict of interest statement in the “Confidential to Editor” section, and submit your "Accept" recommendation.

Reviewer #2: All comments have been addressed

2. Is the manuscript technically sound, and do the data support the conclusions?

Reviewer #2: Yes

3. Has the statistical analysis been performed appropriately and rigorously? 

Reviewer #2: Yes

4. Have the authors made all data underlying the findings in their manuscript fully available?

Reviewer #2: Yes

5. Is the manuscript presented in an intelligible fashion and written in standard English?

Reviewer #2: Yes

6. Review Comments to the Author

Reviewer #2: The authors have addressed all the comments. I have no further comments. However, I would encourage the authors to read the manuscript carefully for typo and grammatical errors. I feel some sentences could be expressed better.

7. PLOS authors have the option to publish the peer review history of their article (what does this mean?). If published, this will include your full peer review and any attached files.

Reviewer #2: No

---

## [Editor Report · Acceptance letter]

19 Feb 2021

PONE-D-20-25237R1 

Reference intervals of common clinical biochemistry analytes in young Nigerian adults 

Dear Dr. Ayemoba:

I'm pleased to inform you that your manuscript has been deemed suitable for publication in PLOS ONE. Congratulations! Your manuscript is now with our production department. 

Kind regards, 

on behalf of

Dr. Ulrike Gertrud Munderloh 

Academic Editor

PLOS ONE